# Basal Bark Treatment of Imidacloprid for Hemlock Woolly Adelgids (*Adelges tsugae*)

**Matthew Quinterno** [1], **Gregory Dahle** [1,*], **Kathryn Gazal** [1], **Anand Persad** [2] and **Jason Hubbart** [1]

[1] Division of Forestry and Natural Resources, Davis College, West Virginia University, Morgantown, WV 26506, USA; mattq412@gmail.com (M.Q.); kathryn.arano@mail.wvu.edu (K.G.); jason.hubbart@mail.wvu.edu (J.H.)

[2] Research, Science & Innovation (RSI), ACRT, 4500 Courthouse Boulevard, Stow, OH 44224, USA; apersad@acrtinc.com

\* Correspondence: gregory.dahle@mail.wvu.edu

**Abstract:** Hemlock wooly adelgid (*Adelgis tsugae* Annand) (HWA) has invaded much of eastern hemlock's (*Tsugae canadensis* L. Carrière) native range. Arborists and forest managers have successfully handled this pest using either contact or systemic pesticides. One of these pesticides, imidacloprid, has often been applied using a soil or trunk injection. Although imidacloprid has been labeled as a basal bark spray to control HWA, minimal information regarding its efficacy is available. This study compared bark treatments to soil treatment of imidacloprid at high and low application rates, as well as the use of a bark adjuvant. The results showed that basal bark treatments were as effective as soil treatments. Hence, basal bark treatments of imidacloprid can be an effective method for control of HWAs in eastern hemlock trees. A bark adjuvant may not be necessary, as it was not found to influence the amount of imidacloprid in the tissues.

**Keywords:** basal bark application; bark sprays; hemlock woolly adelgid; integrated pest management





## 1. Introduction

Eastern hemlock (*Tsugae canadensis* L. Carrière) inhabits much of North America's eastern seaboard forests. Preferring slopes, it can be found from Nova Scotia to Minnesota and south along the Appalachian Mountains to Alabama and Georgia [1]. This tree is an important component of forest riparian systems. For example, riparian eastern hemlock has been shown to heavily influence benthic invertebrate functional feeding group composition in headwater stream communities [2]. Similarly, the consistent addition of woody debris and food resources that eastern hemlock provides could be linked to the abundance of macro-invertebrate shredders in eastern hemlock streams relative to their deciduous counterparts during the summer. This is important, considering that the loss of eastern hemlock due to hemlock woolly adelgid invasion could lead to changes in stream communities and trophic cascades [3]. The value of eastern hemlock in riparian ecosystems is of particular interest, given that previous work demonstrated that no other native evergreen in the Appalachians would likely fill the ecohydrological role of eastern hemlock if widespread mortality were to occur [4]. Eastern hemlock is beneficial as an urban landscape tree, particularly as an individual specimen tree or as group plantings as a screen [1].

Much of eastern hemlock's native range has been affected since the 1950 discovery of hemlock woolly adelgid (*Adelges tsugae* L. Carrière) (HWA) in the United States. DNA evidence has suggested that HWAs arrived from southern Japan and did not migrate from the North American west coast, where this pest is also native [5,6]. HWAs can be found in all ages of tree growth following initial infestation, although soft new growth is the most susceptible [7,8]. Infestation can extend to become stand mortality [9,10]. Stand decline is known to impact forest streams due to both infestation and pre-emptive salvage logging,

either of which can greatly impact the microclimate of hemlock forests and the many associated taxa [11]. Similarly, the accelerated inputs of detritus resulting from hemlock mortality are likely to influence carbon and nutrient fluxes and determine future patterns of species regeneration in these forest ecosystems [12]. Previous researchers have shown that hemlock decline may result in long-term changes in headwater ecosystems, reducing within-stream and entire drainage system benthic community diversity [13]. Webster et al. (2012) [14] provided results that agree with those of [13], showing that contributions of hemlock to litterfall, in-stream wood, and benthic organic matter were significant, suggesting that the loss of hemlock may significantly modify the trophic dynamics and physical structure of the southern Appalachian streams. Previous studies have also shown that streams draining watersheds where eastern hemlock has been lost due to HWA infestation demonstrate permanent reductions in water yield and transient increases in peak flow during large-flow events. Ultimately, the management of riparian forests undergoing hemlock decline should focus on facilitating a faster transition to hardwood-dominated stands to minimize long-term effects on water quality [15] and aquatic biota [13].

Mortality in the urban landscape can lead to loss of amenity or screening value and a direct financial expense for removing a potentially hazardous tree [16]. There are various cultural, chemical, and biological controls for HWAs [17–19]. Pesticide treatments can effectively reduce adelgid populations, improving tree health [17]. However, pesticide treatments in forest stands may be limited to select stands due to the high cost of treating large numbers of trees [17] and label requirements that restrict the amount of chemicals that can be applied per unit of land area ($m^2$ or acre). These constraints are less of a hindrance in the urban landscape, as urban hemlock tree plantings are often managed at the property level.

Imidacloprid is an active tool for managing HWAs as a foliar spar or through systemic application to the trunk or roots. Until recently, soil drenches, soil injections, tablets, and foliar sprays were the approved and preferred methods of applying imidacloprid for HWA control [17]. Soil injections and soil drenches require equipment, such as spray rigs or specialized injectors, while tablets do not require any. Foliar applications require specialized equipment and can induce chemical drift. This fine mist lands off the intended target and perhaps onto adjacent properties. This unnecessarily exposes those environments to chemicals and the applicators to potential liability. Imidacloprid is transformed in eastern hemlock to form the dominant metabolite imidacloprid olefin (henceforth called olefin), which has a toxicity ranging from 10 to 16 times that of the parent compound [19]. Olefin persists within the tree, providing it with a long period of protection against adelgid [19].

Recently, bark applications of imidacloprid have been labeled for HWA. For managers, this method could be less cumbersome and faster for treating HWA. Bark applications may provide several benefits over other application methods. Early research has shown that depending on the product sprayed, bark applications could require as little as 10%–20% of the active ingredients necessary in soil applications to attain the same concentration levels [20,21]. Basal bark applications could reduce translocation time, labor, and material costs and potentially minimize environmental exposure.

The objective of the current study was to compare treatments of imidacloprid as a bark spray to soil application to determine: (1) if bark applications resulted in imidacloprid or olefin concentrations within the trees, (2) whether labeled rates influenced detectable levels of either compound, (3) the necessity of an adjuvant to aid imidacloprid movement into the tree, (4) if bark applications reduce *A. tsugae* populations, and (5) the feasibility of such an application. These comparative results will assist forest managers in making better science-based decisions regarding whether this method can serve as a viable tool in integrated tree pest management.

## 2. Materials and Methods

A field study was conducted at Fallingwater, a Western Pennsylvania Conservancy property in Mill Run, PA USA. The study site was very rocky, with the soils primarily

composed of silt loams. The slopes ranged from 35% to 70% [22]. Fifty-six trees were identified, flagged, and numbered. Eastern hemlocks were selected between 1.5 m and 11.2 m tall and spaced no closer than 9 m. Live crown ratio (LCR) and diameter at breast height (DBH) were measured at this time to gauge whether these cofactors influenced insecticide concentrations.

Once selected, trees were randomly assigned to receive one of seven treatments: six with imidacloprid and a seventh with an untreated control. The treatment trees ranged from 2.4 m to 11.3 m in height and from 6.1 cm to 23.6 cm in DBH. Xytect 75 WSP© (Rainbow Treecare Scientific, Minnetonka, MN, USA, containing 75% imidacloprid) was used for the imidacloprid treatments. Xytect 75 WSP packets at the low rate (0.67 g AI/ 2.5 cm DBH) were mixed with water (1 packet per 11.4 L of water) and two packets per 11.4 L of water for the high rate (1.37 g AI/2.5 cm DBH). The treatments included soil injections at the low versus high rates, bark application at the low versus high rates, and bark application with adjuvant at the low and high rates. The adjuvant was Pentra-bark© (Agbio, Inc. Westminster, CO, USA), and both the imidacloprid and adjuvant were supplied by Rainbow Treecare Scientific. Eight trees were treated in each of the six treatments listed, with a seventh group of eight untreated trees (UTC) used as a control.

Soil injections were made using a soil injector and bottle pump (SPS Systems International, Santa Monica, CA USA), while bark applications were made using a backpack sprayer (Greenwood™ Camarillo, CA, USA). No calibration was required for the bottle pump as the water level inside the container was visible, the volume markings were clearly labeled, and the container was in front of the technician during use. Bark applications were applied with a backpack sprayer that was calibrated by averaging the time it took to fill a 0.24 L container three times. The applications were made on 12–13 May 2017.

## 2.1. HWA Population Counts and Mortality

Three HWA population counts were conducted for each tree. The first was collected in the field on 11 May 2017 (pre-treatment), the second on 20 November 2017 (6 months post-treatment), and the last on 12 April 2018 (11 months post-treatment). The collection was conducted by dividing the canopy into three vertically stratified layers, and, within those layers, four quadrants were set up to form twelve sample units. One 10 cm distal branch tip, representing each sample, was collected using a pole pruner, bagged, labeled, and brought to the lab on ice, where it was stored at 4 °C.

Tallies were made using a stereoscope (Leica Zoom 2000; Leica, Wetzlar, Germany) and probed to elicit HWA movement. When no movements were observed, HWAs were punctured to observe hemolymphs. HWAs were considered dead if they were dry, displayed no movement, or if full of black hemolymphs. Percent mortality was calculated for each group.

## 2.2. Hemlock Tissue Preparation and Analysis

Foliage tissue samples and population samples were collected during the 6-month post-treatment site visit on 20 November 2017. Pole pruners were used to collect four representative samples from the tips of branches growing in the middle quadrant of the canopy. Branches from each tree were pooled in the bagging process and transferred on ice to the lab. Samples were transferred into paper bags, placed under a black bag, and left in the dark to air dry for one week at 23.8 °C. Once dried, needles were ground using a coffee grinder (Mr. Coffee™, Rye, NY, USA). UTC specimens were processed first, followed by bark and soil applications. Isopropyl alcohol (99%) wipes were used to clean the grinder after processing each sample. One gram of the needle grindings was placed into 15 mL centrifuge tubes, labeled, kept under dark/dry conditions, and sent to Villanova University for chemical analysis.

The lab at Villanova University, Villanova, PA, USA, analyzed the needle grindings for imidacloprid and olefin using the liquid chromatography–tandem mass spectrometry (LC/MS/MS) protocol detailed by the authors of [19]. Imidacloprid and olefin quantifi-

cations were conducted with an HPLC system composed of binary Shimadzu LC-20AD pumps and a SIL-20A auto-sampler (Shimadzu, Columbia, MD, USA). Analyst software controlled HPLC separation (Applied BioSystems/SCIEX, Framingham, MA, USA) and ran through a 2 mm Phenomenex Genini NX 2 mm guard column.

### 2.3. Cofactor Measurement and Cost Comparison

To account for variation in concentration levels, diameter at breast height (DBH, measured at 1.4 m), live crown ratio (LCR, length of crown/total height), soil moisture, and precipitation quantities were recorded when the trees were initially selected for this study. Volumetric soil moisture data was measured using a time domain reflectometer (TDR, Model CS605, Campbell Scientific, Logan, UT, USA). Thirteen of the fifty-six treated trees were randomly selected to receive soil moisture readings twice monthly between June and November of 2017. Three readings were taken from inside the dripline of the thirteen trees and then averaged. The daily max, min, and precipitation were recorded from the nearest NOAA station in Confluence, PA USA (~14 km away).

Cost comparisons between soil injections and bark applications were calculated per tree basis. The formula was:

$$\text{Cost} = (\text{USD } 18.55 * \text{application time}) + (\text{USD } 0.13 * \text{imidacloprid}) + (\text{USD } 0.06 * \text{adjuvant})$$

where the cost is denoted in US dollars (USD), the application time is denoted in hours, imidacloprid is in mL, and the adjuvant is also in mL.

The national mean wage of arborists (USD 18.55/hour) was obtained from the United States Department of Labor website [23]. The application time included the time to apply the treatment plus the average time to calibrate and mix each treatment. Imidacloprid was USD 3.86/29 mL, and the adjuvant was USD 1.65/29 mL.

### 2.4. Statistical Analysis

Orthogonal contrasts [24,25] were used to analyze the tissue concentrations, counts of live HWA adults, percent HWA mortality, and financial efficacy. Contrasts were used to reduce the type I error rate. The selected main effect contrasts were: (1). UTC vs. all treated trees, (2). soil treatments vs. bark treatments, (3). low dosage vs. high dosage, and (4). bark treatments vs. bark + adjuvant treatments. The concentration data required a log(x + 1) transformation to obtain a normal distribution of residuals, while the population counts followed a negative binomial distribution; thus, the Laplace method and a log transformation link were deployed. The PROC GLM and GLIMMIX procedures of SAS®were used for the one-way ANOVA analyses (SAS®, Version 9.4, SAS Institute Inc., Cary, NC, USA). The influence of DBH and LCR and each of their interactions with the treatment on imidacloprid and olefin concentrations in leaf tissue were assessed using ANOVA analysis through the PROC GLM feature [26]. The alpha level for all tests was 0.05.

## 3. Results

### 3.1. Tissue Analysis

All insecticide treatments produced detectable levels of imidacloprid and olefin in leaf tissues with significantly higher log(x + 1) concentrations of imidacloprid ($p < 0.0001$. N = 56) and olefin ($p = 0.0376$, N = 56) than the untreated control trees (Table 1). The labeled dosage did not significantly affect the mean log(x + 1) concentrations of imidacloprid ($p = 0.1181$, N = 48) and olefin ($p = 0.5013$, N = 48). The adjuvant did not affect the mean log(x + 1) concentrations of imidacloprid ($p = 0.8358$, N = 32) or olefin ($p = 0.7721$, N = 32). No significant differences were found between the bark treatments and the soil treatments on the log(x + 1) concentrations of imidacloprid ($p = 0.0789$, N = 48) or olefin ($p = 0.2121$, N = 48).

**Table 1.** One-way ANOVA with orthogonal contrasts of the mean concentration $\pm$ SE (standard error of the mean) testing the effect of imidacloprid and olefin concentrations six months post-treatment. Statistical analysis was conducted on log(x + 1)-transformed data, and for practical purposes presented as original data.

| Treatment | N | Imidacloprid (ppb) | | Olefin (ppb) | |
|---|---|---|---|---|---|
| | | Mean $\pm$ SE | *p*-Value | Mean $\pm$ SE | *p*-Value |
| Control | 8 | 0.00 $\pm$ 0.00 | <0.0001 | 0.00 $\pm$ 0.00 | 0.0376 |
| Treated | 48 | 85.12 $\pm$ 24.33 | | 10.64 $\pm$ 2.93 | |
| Low dose | 24 | 91.18 $\pm$ 45.20 | 0.1181 | 9.92 $\pm$ 4.55 | 0.5013 |
| High dose | 24 | 79.06 $\pm$ 19.34 | | 11.37 $\pm$ 3.80 | |
| Soil | 16 | 161.96 $\pm$ 68.55 | 0.0789 | 19.69 $\pm$ 7.78 | 0.2121 |
| Bark | 32 | 46.70 $\pm$ 8.20 | | 6.12 $\pm$ 1.71 | |
| Bark | 16 | 44.16 $\pm$ 10.46 | 0.8358 | 5.59 $\pm$ 2.18 | 0.7721 |
| Bark + adjuvant | 16 | 49.26 $\pm$ 12.94 | | 6.65 $\pm$ 2.69 | |

*3.2. Biological Efficacy*

No differences ($p$ = 0.8799, N = 56) were found in the presence of live adult HWAs at the time of treatment (spring 2017) between the control trees ($\bar{x}$= 8.6 $\pm$ 4.9 SE, n = 8) and the treated trees ($\bar{x}$ = 15.1 $\pm$ 5.7 SE, n = 48). The mean log number of live adult HWAs six months post-treatment (fall 2017) was significantly lower ($p$ = 0.0155, N = 56, Table 2) in the treated trees when contrasted with the control trees. Furthermore, a significant difference was found in the six months post-treatment ($p$ = 0.0216, N = 48) between the bark treatments and the soil treatments, yet no differences were found between the low-dose level vs. high-dose level ($p$ = 0.2428, N = 48) or bark vs. bark + adjuvant ($p$ = 0.9755, N = 32). At eleven months post-treatment, there were no differences between the live adult HWAs for the control vs. treated trees ($p$ = 0.9817, N = 56), soil vs. bark treatments ($p$ = 0.9778, N = 48), high dose vs. low dose ($p$ = 0.9861, N = 48) or bark vs bark + adjuvant ($p$ = 0.9747, N = 32).

**Table 2.** One-way ANOVA with selected orthogonal contrasts of mean log live HWA adults for six months post-treatment and eleven months post-treatment. Statistical analysis was conducted on log-transformed data. Means and standard errors (SEs) are presented as data before transformation.

| Treatment | N | Six Months Post-Treatment (Fall) | | Eleven Months Post-Treatment (Spring) | |
|---|---|---|---|---|---|
| | | Mean $\pm$ SE | *p*-Value | Mean $\pm$ SE | *p*-Value |
| Control | 8 | 99.1 $\pm$ 45.2 | 0.0155 | 0.6 $\pm$ 0.3 | 0.9817 |
| Treated | 48 | 19.8 $\pm$ 5.5 | | 2.3 $\pm$ 1.3 | |
| Low dose | 24 | 23.0 $\pm$ 7.8 | 0.2428 | 4.0 $\pm$ 2.6 | 0.9861 |
| High dose | 24 | 16.0 $\pm$ 7.8 | | 0.7 $\pm$ 0.5 | |
| Soil | 16 | 36.3 $\pm$ 11.9 | 0.0216 | 1.1 $\pm$ 0.7 | 0.9778 |
| Bark | 32 | 9.9 $\pm$ 4.3 | | 3.1 $\pm$ 2.1 | |
| Bark | 16 | 8.9 $\pm$ 3.8 | 0.9455 | 0.1 $\pm$ 0.1 | 0.9747 |
| Bark + adjuvant | 16 | 10.8 $\pm$ 7.5 | | 5.6 $\pm$ 3.8 | |

The HWA log percent mortality at six months post-treatment was higher ($p < 0.0001$, N = 56, Table 3) in the treated trees relative to the control trees as well as at 11 months post-treatment ($p = 0.0025$, N= 56). The only treatment type that differed in terms of log mortality was the bark treatments ($\bar{x}$ = 91.6 ± 3.9, n = 32), which had a higher mortality than the soil treatments ($\bar{x}$= 57.6 ± 0.2, n = 16) at six months ($p = 0.0384$, N = 48). All other treatments were not found to differ at six months or eleven months post-treatment.

**Table 3.** One-way ANOVA with orthogonal contrasts of mean log HWA percent mortality for six months post-treatment and eleven months post-treatment. Statistical analysis was conducted on log-transformed data. Means and standard errors (SEs) are presented as data before transformation.

| Treatment | N | Six Months Post-Treatment (Fall) | | Eleven Months Post-Treatment (Spring) | |
|---|---|---|---|---|---|
| | | Mean ± SE | *p*-Value | Mean ± SE | *p*-Value |
| Control | 8 | 20.7 ± 0.2 | <0.0001 | 37.5 ± 12.5 | 0.0025 |
| Treated | 48 | 80.3 ± 6.9 | | 81.0 ± 7.8 | |
| | | | | | |
| Low dose | 24 | 77.8 ± 8.4 | 0.7490 | 65.7 ± 14.8 | 0.2365 |
| High dose | 24 | 82.8 ± 3.5 | | 96.3 ± 2.4 | |
| | | | | | |
| Soil | 16 | 57.6 ± 0.2 | 0.0384 | 48.8 ± 21.3 | 0.0855 |
| Bark | 32 | 91.6 ± 3.9 | | 97.1 ± 1.2 | |
| | | | | | |
| Bark | 16 | 94.2 ± 2.8 | 0.7803 | 99.6 ± 0.3 | 0.8987 |
| Bark + adjuvant | 16 | 89.0 ± 7.4 | | 94.4 ± 2.3 | |

### 3.3. Cofactors and Cost Comparison

Soil moisture was not limiting across the site for plant growth in the silt loams. In the silt loams, water is plant available when its percentages are above 10%–15% [27], and all measurements were at or above this level (Table 4). As such, water uptake and imidacloprid translocation should not have been inhibited. The monthly minimum temperatures ranged from 7.8 °C to −24.4 °C between the winter months of November and March. There were seven consecutive nights in January with minimum temperatures between −19.4 °C and −24.4 °C.

**Table 4.** Mean percent soil moisture.

| Month | Percent Soil Moisture (%) |
|---|---|
| June | 19.6 |
| July | 22.3 |
| August | 16.8 |
| September | 14.0 |
| October | 23.9 |
| November | 23.7 |

The ANOVA statistical test did not reveal a relationship between the live crown ratio and the effect of treatment (six pesticide treatments) on the tissue concentration of either compound tested: imidacloprid ($p = 0.2068$, N = 48) or olefin ($p = 0.1224$, N = 48). DBH did not affect the concentrations of olefin ($p = 0.1580$, N = 48), yet DBH was found to affect the tissue concentrations of imidacloprid (IC) ($p = 0.0167$, $R_2 = 0.4452$). Specifically, there was a negative slope when the concentration of imidacloprid was regressed on DBH across

all pesticide-treated groups. However, there was no interaction between the LCR or DBH and the treatments, indicating a similar slope for each of the six treatments. The individual regression equations for all six treatments were as follows:

- Soil-drench low: IC = 4.88 − 0.06 ∗ DBH;
- Soil high: IC = 7.34 − 0.22 ∗ DBH;
- Bark low: IC = 6.34 − 0.24 ∗ DBH;
- Bark high: IC = 6.02 − 0.24 ∗ DBH;
- Bark low + adjuvant: IC = 5.71 − 0.24 ∗ DBH;
- Bark high + adjuvant: IC = 4.24 + 0.0004 ∗ DBH.

The high doses were more expensive ($p = 0.0004$, N = 24, Table 5) than the low doses in this study. No other treatment was found to differ: bark vs. soil ($p = 0.1194$, N = 48) or bark versus bark + adjuvant ($p = 0.1009$, N = 32).

**Table 5.** One-way ANOVA with orthogonal contrasts for treatment costs.

| Comparison | N | Mean Cost (USD) ± SE | *p*-Value |
|---|---|---|---|
| Control | 8 | 0.00 ± 0.00 | 0.0015 |
| Treated | 48 | 1.79 ± 0.07 | |
| | | | |
| Soil | 16 | 1.77 ± 0.15 | 0.1194 |
| Bark | 32 | 1.80 ± 0.08 | |
| | | | |
| Low | 24 | 1.55 ± 0.09 | 0.0004 |
| High dosage | 24 | 2.04 ± 0.10 | |
| | | | |
| Bark | 16 | 1.68 ± 0.10 | 0.1009 |
| Bark + Pentra-bark | 16 | 1.91 ± 0.13 | |

## 4. Discussion

This short-term assessment of the efficacy of imidacloprid basal bark applications demonstrated that these treatments can be effective for hemlock woolly adelgid control in eastern hemlocks. This study found that imidacloprid and olefin were recovered in leaf tissues after basal bark applications, and that their concentrations were similar to those associated with soil injections. In addition, concentrations of imidacloprid and olefin were similar to those found in other studies, despite different application methods [19,28,29]. In one instance, where bark applications were also used, the concentrations of imidacloprid obtained in this study appeared higher than that retrieved by McCullough et al. (2011) [30]. The results of the current study demonstrated that bark penetration was not influenced by the labeled dosage or the use of an adjuvant. These findings also suggested that HWA mortality via bark application is comparable to that via soil injection, and that bark applications may produce higher mortality levels in a shorter period of time.

The addition of an adjuvant in this study was not found to increase the concentrations of imidacloprid or olefin nor lead to greater HWA mortality. These results corroborate the findings of both Cowles (2010) [31] and McCullough et al. (2011) [30]. While the cost of using an adjuvant was not significantly higher, there appears to be little statistical support to justify using an adjuvant when applying imidacloprid basal bark applications for HWAs.

The mortality rate was significantly higher in the bark treatments over the soil treatments six months post-treatment but did not differ eleven months post-treatment. This difference was found despite the lack of a significant difference in the concentrations of imidacloprid or olefin in the tissue samples taken in the fall. The quick mortality via basal bark applications was unexpected. However, it was assumed that the insecticide entered the xylem more quickly, as it is not as dependent on soil moisture for its uptake and trans-

port [20]. These results vary slightly from previous research conducted by Faulkenberry (2012) [32], who showed that the mortality rate between imidacloprid bark applications and soil drenches at approximately six months post-treatment did not significantly differ (CI: 0.05). Future research should investigate whether imidacloprid bark applications move more quickly into the xylem by sampling tissues throughout the growing season. The HWAs' cold hardiness diminishes over the cold months as March approaches, and prolonged cold temperatures of $-20\ °C$ can impact HWA populations [33]. The monthly mean minimum temperature in the study area reached its lowest in January, with an average of $-11\ °C$. However, as there were seven consecutive nights with minimum temperatures between $-19.4\ °C$ and $-24.4\ °C$, these cold temperatures could have contributed to increased mortality across the treatments.

While the cost for the bark applications of imidacloprid was not found to differ from that of the soil applications, having the bark application in the IPM toolbox permits additional flexibility. Additionally, since there was no difference between the high and low rates of imidacloprid concentration in leaf tissues and HWA mortality, using a lower concentration could reduce costs by approximately 32%. However, the trees in this study were not large, so these results have only been demonstrated in trees less than 24 cm in diameter.

The results of this study support the need for future studies for HWA control using basal bark applications. Researchers may wish to assess treatments on stands with a higher HWA population. The trees in this study had a relatively low level of HWA infestation. With such a high natural mortality in the untreated control and a small number of living insects, it may be advisable to research stands with a denser population to obtain a clearer picture of how this application method affects pest populations. Second, the long-term effects of bark treatments regarding longevity, tissue concentration, and HWA mortality should be studied.

## 5. Conclusions

This study demonstrated that basal bark treatments of imidacloprid can be an effective method for control of HWAs in eastern hemlock trees. When designing treatment regimes, practitioners may select the lower labeled dose for trees less than 24 cm in diameter, as there was no difference in the mortality of HWAs between application rates. Furthermore, using a bark adjuvant may not be necessary, as it was not found to influence the amount of imidacloprid in the tissues.

**Author Contributions:** Conceptualization, M.Q.; formal analysis, M.Q. and G.D.; funding acquisition, M.Q., G.D. and A.P.; methodology, M.Q., G.D. and A.P.; writing—original draft, M.Q., G.D., K.G., J.H. and A.P. All authors have read and agreed to the published version of the manuscript.

**Funding:** This project was partly funded by the West Virginia University Division of Forestry & Natural Resources, NIFA McIntire-Stennis Grants WVA00108, WVA00813 and 7003934, WVU West Virginia Agriculture and Forestry Experiment Station, and the Davey Institute. Rainbow Treecare Scientific donated product (Xytect 75 WSP).

**Data Availability Statement:** Data are contained within the article.

**Acknowledgments:** We would like to thank Elizabeth McCarthy, Rick Turcotte, Anthony Lagalante, and Richard Cowles for providing their insight in project development. The Lagalante lab at Villanova University conducted the chemical analysis for this project. John Seifer of the Davey Institute assisted in the fieldwork, and Ida Holásková assisted with the statistical analyses.

**Conflicts of Interest:** The authors declare no conflict of interest. The funders had no role in the design of the study; in the collection, analyses, or interpretation of data; in the writing of the manuscript; or in the decision to publish the results.

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
