# Peer review of "Basal Bark Treatment of Imidacloprid for Hemlock Woolly Adelgids (Adelges tsugae)"

_forests, doi:10.3390/f14112229_

Round 1
Reviewer 1 Report
Comments and Suggestions for Authors
The problem is well presented as well as obtained results.
Author Response
We thank the reviewers for providing valuable recommendations. Our revised manuscript reflects the recommended changes. We used track changes to highlight our additions. In this revision, we have updated the in-text citations to reflect the journal's preference for numerical listing of citations. We did not highlight these changes with track changes as it would be cumbersome to read the document.
We address specific comments below.
Reviewer 1
The problem is well presented as well as obtained results.
Response: Thank you
Reviewer 2 Report
Comments and Suggestions for Authors
The paper “Basal bark treatment ….”, by Quinterno, et al., is a valuable contribution to the field of managing hemlock woolly adelgids in forest and managed landscapes. The manuscript requires rewriting for it to be considered acceptable. I am concerned that the manuscript was submitted prematurely, without all the revisions from all the coauthors. Otherwise, how is it possible that four coauthors didn’t notice that the References cited weren’t even organized in alphabetical order? Any member of a graduate student’s committee and with their name on a manuscript should have assisted the student in making sure that there were no further changes required to the manuscript – before submitting the paper for outside peer review. Frankly, either the statement under Author Contributions that all five authors worked on the manuscript and found it to be acceptable for submission is beyond belief. Outside reviewers should not end up doing the work of the graduate committee. Overall, this manuscript appears more to be an early draft than one suitable for being submitted.
Major problems with the manuscript
11 Alphabetize the references and use a standardized style acceptable to MDPI journals for citing the literature. It is very unusual to present the title of journal articles in initial capitals: e.g., Adkins et al. 2015. Rather, the style should be more like that used for the Snyder et al. 2002 reference.
22 The analytical chemistry was conducted at Villanova University, which suggests that Dr. Anthony Lagalante was involved with processing and analyzing these samples. Why is he not a coauthor? He is mentioned in the acknowledgements as having provided insight into project development but doesn’t mention that he analyzed the pesticide residues. Rather than struggling with describing (poorly*) and repeating the standardized methods used in the Lagalante lab, the analytical methods can simply be referenced by citing Benton et al. 2015, together with a brief outline of the process: e.g., “One gram of pulverized dried tissue from each sample was extracted with acetonitrile and analyzed via HPLC/MS/MS via the methods of Benton et al. 2015.” If there were any changes to the equipment or protocol in the lab, then these changes should be noted.
Benton, E.P., Grant, J.F., Webster, R.J., Nichols, R.J., Cowles, R.S., Lagalante, A.F., Coots, C.I., 2015. Assessment of imidacloprid and its metabolites in foliage of eastern hemlock multiple years following treatment for hemlock woolly adelgid, Adelges tsugae (Hemiptera: Adelgidae), in forested conditions. J. Econ. Entomol. http://dx.doi.org/10.1093/jee/tov241
*The “poorly” comment is with respect to spelling (concertation, Line 188), missing characters for micro on lines 177 and 188. The comment about standards (Lines 180 – 181) doesn’t add any information to the paper. What were the standards, what were their sources, and what range of concentrations provided a linear response from the instrument? Were there internal standards used in the quantitation? It is stated that the LOD was calculated – what were these concentrations for imidacloprid and the imidacloprid olefin?
33. The section “Cofactor Measurement and Cost Comparison” contains a description of efforts to measure precipitation and soil moisture around representative trees, with the objective to correlate these data with tissue concentration differences of the applied pesticide. While it is good that soil moisture data were recorded, to show that the experiment didn’t take place during drought conditions, the idea that these data are useful for correlation or regression analyses is misguided. The study area had to have been too small for there to be much difference among trees with respect to precipitation amounts. It is noted that the nearest weather station recording precipitation was ~14 km distant, which means that there could be no way to use these data to “account for variation in mortality rates” (Lines 201 – 202). The entire section needs to be reworked to match how the data are being used.
44 The experiments appeared to have been based on a completely randomized design with 8 replicates (the experimental design should be explicitly stated). This may have been problematic. The study trees varied from 1.5 to 11.2 meters in height. That is a considerable range in size. A range in the tree DBH used in this study should also be stated, as this is the measure that was used for calculating dosage. If not blocked by height or DBH, then the variation in DBH that was noted to have influenced imidacloprid concentration in plant tissues (Lines 284 – 295) could have been unevenly represented with the various treatments. Thus, the differences measured may be due to a poor randomization protocol, rather than true treatment differences. Just a side question: what is the DBH for a 1.5 meter tall tree? The range of DBH could be given for the different treatment groups to clear up the possible problem of confounding.
55 There is a sloppiness in terminology used in this manuscript with respect to mortality. Line 220 mentions “mortality rate”. Does the author really mean percent mortality? The two are not the same - rate commonly means something with elapsed time in the denominator. Table 4 is currently uninterpretable and may have used inappropriate transformations for analyzing the data. “Mean log HWA mortality” has not been defined. We know that live and dead HWA individuals were assessed from individual branch samples. From this, the percent mortality of each sample was probably calculated. If that was the variable, subjecting a percent value to a log transformation is unusual and probably inappropriate. There are many transformations that are useful for normalizing the variance of percent data, such as probit, logit, or the normsinv function in Excel.
66 On the subject of statistics: P-values should not be reported to four significant digits, two will suffice. There is no value to reporting P values of 0.9817, 0.9861, 0.9771 amd 0.9747! As far as statistical significance is concerned, reporting P > 0.5 would be just as informative.
For reporting means with some measure of dispersion, at the first occasion, mention what measure of dispersion is being reported (e.g. Line 248 would be “… between the control trees (8.6 +/- 4.9, mean +/- s.d.) and the …”.
Reporting back-transformed means following log transformation is fine, but a problem creeps in when presenting the standard deviations, standard errors, or confidence limits. These values are asymmetric around the mean following back-transformation. To be precise, the lower and upper limits of these values must be given, rather than a single value.
When describing log transformation, use the following style “data required a log(x+1) transformation”. This is necessary, as the log of zero is undefined, and I am sure you had plenty of zero values.
77. Table 1 is unnecessary. The entire experiment can be stated as follows: “An experiment was designed to compare soil injection vs. bark spray application of imidacloprid at a low vs. high dosage (0.67 vs. 1.37 g active ingredient per 2.5 cm diameter dbh), with a pair of additional bark spray treatments assessing the same dosages applied with an adjuvant. These treatments were compared with each other and with an untreated control group in a completely randomized block experiment with eight replicates.” Additional details would then be given regarding the precise methods of applying the treatments. Note that the methods are currently incomplete, as the volume of water being applied with the products is not described. This is an important detail that must be included!
An alternative to the method used above would be to recognize that the experiment was designed as a 3 x 2 factorial design, plus an untreated check. However, the data were not analyzed as a factorial experiment, and so method would mislead the readers.
88. The abstract and conclusion state that the low rate tested in this work is adequate for treating trees. This is an incorrect extrapolation of the results beyond the experimental population. Trees in the study only varied from 1.5 to 10 m in height, which probably only represented trees up to about 20 cm DBH. The work by Benton and others included trees of a much greater range in DBH and the relationship between DBH and effective dosages is well established, from a much larger population and with repeated samples to determine insecticide residues and treatment effects on HWA over several years. The statement that practitioners should choose the lower dosage should be removed from the abstract and the conclusion, unless it is conditioned by the statement that this has been demonstrated to be effective for trees up to x cm DBH.
Petty annoyances that need correcting:
Be consistent throughout the paper with spelling and capitalization. Follow standard rules. The title of the paper disagrees in spelling with the abstract! My preference is hemlock woolly adelgid (two Ls in woolly). Common names are not capitalized unless they include a proper name (e.g. hemlock woolly adelgid vs. Fletcher scale). This also applies to common names of chemicals: imidacloprid is correct, but the trade name is Xytect 75 WSP.
Line 48. Change to “… infestation can expand to become stand mortality …”
Line 71 and Line 440. Correct spelling to “Havill”.
Line 74 – 76. Dirr 1998 mentioned nothing about landscape populations being less prone to HWA population spread, so this reference is irrelevant. I would disagree with the statement, anyway. Eastern hemlock are often planted as hedges, and this results in a greater plant density than is commonly found in forests.
Lines 77 – 79. Reword. “Imidacloprid is active for managing HWA either as a foliar spray or through systemic application to the trunk or roots.” The next sentence, Lines 78 – 80 is a mess of verbiage, some of which is irrelevant (low vapor pressure has little significance to systematicity), and should be deleted. The dissociation constant of about 11.1 means that imidacloprid exists as a cation when being transported in trees, which implies that it principally only moves in xylem.
Line 85. Reword. “… how high a spray can reach. Foliar sprays create chemical drift, in which fine mist …”
Lines 87 – 88. This is a misleading statement. The transformation of imidacloprid isn’t just a matter of it being in solution. The chemical reactions mentioned hardly occur in solution in the dark and a neutral pH. A better statement would be “Imidacloprid is transformed in eastern hemlock to form the dominant metabolite imidacloprid olefin (henceforth called “olefin”) which has a toxicity of 10 – 16 times that of the parent compound.” Benton’s work can be cited. The phrase in line 90 “is believed to” must be removed. It is a disservice to Dr. Benton (now McCarty) that the credibility of this established fact is lessened by the authors.
Line 103. The species name Adelges tsugae should be italicized.
Lines 143 – 144. By definition there are at most four quadrants. Reword to “… canopy into three vertically stratified layers and within those layers four quadrants to form 12 sample units. One 10-cm distal branch tip was collected with a pole pruner to represent each sample.” This approach makes me wonder how the samples were collected from the 1.5-m tall tree.
Line 145 – 146. Remove the phrase “in the fridge”
Line 276. Provide a reference for the statement that water is available when present above 14% by weight in soil.
Lines 324 – 329. This section provides a garbled non-explanation of the relationship between DBH and tissue concentrations of systemics. Smaller trees have a smaller biomass of foliage relative to DBH (this is well known as the allometry for each species of tree). When a given amount of active ingredient is applied per unit of DBH, then its dilution into a smaller biomass in a small tree implies a higher resulting concentration in its foliage. The volume of water used by the different sized trees may not have any particular significance with respect to the resulting concentration in tissues.
Line 349 – 351. Delete this sentence, or condition it by stating that it applies only to trees of the size represented by this small study.
Comments on the Quality of English LanguageThe quality of the English is fine, but the grammatical errors (mostly related to capitalization), spelling ("Havill"), and formatting (scientific names should be italicized) need to be corrected.
Author Response
We thank the reviewers for providing valuable recommendations. Our revised manuscript reflects the recommended changes. We used track changes to highlight our additions. In this revision, we have updated the in-text citations to reflect the journal's preference for numerical listing of citations. We did not highlight these changes with track changes as it would be cumbersome to read the document.
We address specific comments below.
REVIEWER 2
Comments and Suggestions for Authors
The paper “Basal bark treatment ….”, by Quinterno, et al., is a valuable contribution to the field of managing hemlock woolly adelgids in forest and managed landscapes. The manuscript requires rewriting for it to be considered acceptable. I am concerned that the manuscript was submitted prematurely, without all the revisions from all the coauthors. Otherwise, how is it possible that four coauthors didn’t notice that the References cited weren’t even organized in alphabetical order? Any member of a graduate student’s committee and with their name on a manuscript should have assisted the student in making sure that there were no further changes required to the manuscript – before submitting the paper for outside peer review. Frankly, either the statement under Author Contributions that all five authors worked on the manuscript and found it to be acceptable for submission is beyond belief. Outside reviewers should not end up doing the work of the graduate committee. Overall, this manuscript appears more to be an early draft than one suitable for being submitted.
Response: Thank you for your review. Regarding the order of the References. We ordered the citation numerically, instead of alphabetically as this is the standard for this journal. We acknowledge that we did not number the in-text citation in our original submission or in the reference list. We have found that when new citations are added during the review process, it makes re-numbering the whole document challenging. In this revision, we have replaced the in-text citation with the corresponding number for the list of references.
Major problems with the manuscript
1 Alphabetize the references and use a standardized style acceptable to MDPI journals for citing the literature. It is very unusual to present the title of journal articles in initial capitals: e.g., Adkins et al. 2015. Rather, the style should be more like that used for the Snyder et al. 2002 reference.
Response: We have replaced the in-text citation with the corresponding number from or the list of references following the standard for the journal in citing references (i.e., numerical ordering as cited in the manuscript rather than alphabetical).
2 The analytical chemistry was conducted at Villanova University, which suggests that Dr. Anthony Lagalante was involved with processing and analyzing these samples. Why is he not a coauthor? He is mentioned in the acknowledgements as having provided insight into project development but doesn’t mention that he analyzed the pesticide residues. Rather than struggling with describing (poorly*) and repeating the standardized methods used in the Lagalante lab, the analytical methods can simply be referenced by citing Benton et al. 2015, together with a brief outline of the process: e.g., “One gram of pulverized dried tissue from each sample was extracted with acetonitrile and analyzed via HPLC/MS/MS via the methods of Benton et al. 2015.” If there were any changes to the equipment or protocol in the lab, then these changes should be noted.
Response: Dr. Lagalante’s was not an author of this manuscript; rather, his lab was contracted to conduct the analytical analysis for this study. As such, we do not feel it would be appropriate for him to be part of the authorship, and this is why his contributions are noted in the methodology and acknowledgement sections. We have re-written this section of the methodology referencing the Bentan et al. 2015 document.
*The “poorly” comment is with respect to spelling (concertation, Line 188), missing characters for micro on lines 177 and 188. The comment about standards (Lines 180 – 181) doesn’t add any information to the paper. What were the standards, what were their sources, and what range of concentrations provided a linear response from the instrument? Were there internal standards used in the quantitation? It is stated that the LOD was calculated – what were these concentrations for imidacloprid and the imidacloprid olefin?
Response: We have re-written this section of the methodology referencing the Bentan et al. 2015 document and also addressed misspellings.
- The section “Cofactor Measurement and Cost Comparison” contains a description of efforts to measure precipitation and soil moisture around representative trees, with the objective to correlate these data with tissue concentration differences of the applied pesticide. While it is good that soil moisture data were recorded, to show that the experiment didn’t take place during drought conditions, the idea that these data are useful for correlation or regression analyses is misguided. The study area had to have been too small for there to be much difference among trees with respect to precipitation amounts. It is noted that the nearest weather station recording precipitation was ~14 km distant, which means that there could be no way to use these data to “account for variation in mortality rates” (Lines 201 – 202). The entire section needs to be reworked to match how the data are being used.
Response: We agree that having a closer weather station would have been preferable. However, that was fiscally infeasible. The data we acquired were relevant as they are partnered with the on-site soil moisture readings. Together, the data demonstrates that there was adequate moisture to support transpiration and, therefore, the transport of the pesticide. We have deleted the comment about mortality rates.
4 The experiments appeared to have been based on a completely randomized design with 8 replicates (the experimental design should be explicitly stated). This may have been problematic. The study trees varied from 1.5 to 11.2 meters in height. That is a considerable range in size. A range in the tree DBH used in this study should also be stated, as this is the measure that was used for calculating dosage. If not blocked by height or DBH, then the variation in DBH that was noted to have influenced imidacloprid concentration in plant tissues (Lines 284 – 295) could have been unevenly represented with the various treatments. Thus, the differences measured may be due to a poor randomization protocol, rather than true treatment differences. Just a side question: what is the DBH for a 1.5 meter tall tree? The range of DBH could be given for the different treatment groups to clear up the possible problem of confounding.
Response: The reference to 1.5 meters is the minimum height for inclusion in the potential pool of trees and is not referring to the size of trees that were sampled. We have added a statement defining the height and diameter range for the trees sampled in this study. This can be found in the second paragraph of the Method section.
5 There is a sloppiness in terminology used in this manuscript with respect to mortality. Line 220 mentions “mortality rate”. Does the author really mean percent mortality? The two are not the same - rate commonly means something with elapsed time in the denominator. Table 4 is currently uninterpretable and may have used inappropriate transformations for analyzing the data. “Mean log HWA mortality” has not been defined. We know that live and dead HWA individuals were assessed from individual branch samples. From this, the percent mortality of each sample was probably calculated. If that was the variable, subjecting a percent value to a log transformation is unusual and probably inappropriate. There are many transformations that are useful for normalizing the variance of percent data, such as probit, logit, or the normsinv function in Excel.
Response: We meant percent mortality and apologize for the sloppiness. All references to mortality rates have been edited. We have edited “Mean log HWA mortality” to read as “Mean log percent mortality”, in which percent mortality is now defined. We worked with a statistician for all analyses who determined that the transformation was appropriate.
6 On the subject of statistics: P-values should not be reported to four significant digits, two will suffice. There is no value to reporting P values of 0.9817, 0.9861, 0.9771 amd 0.9747! As far as statistical significance is concerned, reporting P > 0.5 would be just as informative.
Response: While we agree that reporting P > 0.5 provides knowledge that the results were insignificant, we feel that proving the full p-value adds additional quantitative information and have chosen to report full p-values.
For reporting means with some measure of dispersion, at the first occasion, mention what measure of dispersion is being reported (e.g. Line 248 would be “… between the control trees (8.6 +/- 4.9, mean +/- s.d.) and the …”.
Response: added, thank you
Reporting back-transformed means following log transformation is fine, but a problem creeps in when presenting the standard deviations, standard errors, or confidence limits. These values are asymmetric around the mean following back-transformation. To be precise, the lower and upper limits of these values must be given, rather than a single value.
Response: We have double checked with our statistician. While the statistical analyses were conducted on the transformed data, the means and standard errors that we report were derived from the original (untransformed) data. We feel this makes the presentation of the data easier to read.
When describing log transformation, use the following style “data required a log(x+1) transformation”. This is necessary, as the log of zero is undefined, and I am sure you had plenty of zero values.
Response: Added
- Table 1 is unnecessary. The entire experiment can be stated as follows: “An experiment was designed to compare soil injection vs. bark spray application of imidacloprid at a low vs. high dosage (0.67 vs. 1.37 g active ingredient per 2.5 cm diameter dbh), with a pair of additional bark spray treatments assessing the same dosages applied with an adjuvant. These treatments were compared with each other and with an untreated control group in a completely randomized block experiment with eight replicates.” Additional details would then be given regarding the precise methods of applying the treatments. Note that the methods are currently incomplete, as the volume of water being applied with the products is not described. This is an important detail that must be included!
Response: Table 1 has been deleted, and we added text describing it, and we have re-worded this section of the Methods to simplify and added the amount of Xytect per water
An alternative to the method used above would be to recognize that the experiment was designed as a 3 x 2 factorial design, plus an untreated check. However, the data were not analyzed as a factorial experiment, and so method would mislead the readers.
Response: Please see above
- The abstract and conclusion state that the low rate tested in this work is adequate for treating trees. This is an incorrect extrapolation of the results beyond the experimental population. Trees in the study only varied from 1.5 to 10 m in height, which probably only represented trees up to about 20 cm DBH. The work by Benton and others included trees of a much greater range in DBH and the relationship between DBH and effective dosages is well established, from a much larger population and with repeated samples to determine insecticide residues and treatment effects on HWA over several years. The statement that practitioners should choose the lower dosage should be removed from the abstract and the conclusion, unless it is conditioned by the statement that this has been demonstrated to be effective for trees up to x cm DBH.
Response: We removed the statement pertaining to low dose from the abstract and added a qualifier in the Discussion and Conclusion sections.
Petty annoyances that need correcting:
Be consistent throughout the paper with spelling and capitalization. Follow standard rules. The title of the paper disagrees in spelling with the abstract! My preference is hemlock woolly adelgid (two Ls in woolly). Common names are not capitalized unless they include a proper name (e.g. hemlock woolly adelgid vs. Fletcher scale). This also applies to common names of chemicals: imidacloprid is correct, but the trade name is Xytect 75 WSP.
Response: changed
Line 48. Change to “… infestation can expand to become stand mortality …”
Response: changed
Line 71 and Line 440. Correct spelling to “Havill”.
Response: change and thanks
Line 74 – 76. Dirr 1998 mentioned nothing about landscape populations being less prone to HWA population spread, so this reference is irrelevant. I would disagree with the statement, anyway. Eastern hemlock are often planted as hedges, and this results in a greater plant density than is commonly found in forests.
Response: Thank you, we have removed this citation and reworded the sentence to read: “These constraints are less of a hindrance in the urban landscape, as urban hemlock tree planting are often may be managed at the property level.”
Lines 77 – 79. Reword. “Imidacloprid is active for managing HWA either as a foliar spray or through systemic application to the trunk or roots.” The next sentence, Lines 78 – 80 is a mess of verbiage, some of which is irrelevant (low vapor pressure has little significance to systematicity), and should be deleted. The dissociation constant of about 11.1 means that imidacloprid exists as a cation when being transported in trees, which implies that it principally only moves in xylem.
Response: We edited the first sentence and deleted the second sentence.
Line 85. Reword. “… how high a spray can reach. Foliar sprays create chemical drift, in which fine mist …”
Response: edited
Lines 87 – 88. This is a misleading statement. The transformation of imidacloprid isn’t just a matter of it being in solution. The chemical reactions mentioned hardly occur in solution in the dark and a neutral pH. A better statement would be “Imidacloprid is transformed in eastern hemlock to form the dominant metabolite imidacloprid olefin (henceforth called “olefin”) which has a toxicity of 10 – 16 times that of the parent compound.” Benton’s work can be cited. The phrase in line 90 “is believed to” must be removed. It is a disservice to Dr. Benton (now McCarty) that the credibility of this established fact is lessened by the authors.
Response: Edited
Line 103. The species name Adelges tsugae should be italicized.
Response: Thank you, this has been corrected
Lines 143 – 144. By definition there are at most four quadrants. Reword to “… canopy into three vertically stratified layers and within those layers four quadrants to form 12 sample units. One 10-cm distal branch tip was collected with a pole pruner to represent each sample.” This approach makes me wonder how the samples were collected from the 1.5-m tall tree.
Response: edited
Line 145 – 146. Remove the phrase “in the fridge”
Response: deleted
Line 276. Provide a reference for the statement that water is available when present above 14% by weight in soil.
Response: citation added
Lines 324 – 329. This section provides a garbled non-explanation of the relationship between DBH and tissue concentrations of systemics. Smaller trees have a smaller biomass of foliage relative to DBH (this is well known as the allometry for each species of tree). When a given amount of active ingredient is applied per unit of DBH, then its dilution into a smaller biomass in a small tree implies a higher resulting concentration in its foliage. The volume of water used by the different sized trees may not have any particular significance with respect to the resulting concentration in tissues.
Response: We deleted this paragraph
Line 349 – 351. Delete this sentence, or condition it by stating that it applies only to trees of the size represented by this small study.
Response: Deleted
Comments on the Quality of English Language
The quality of the English is fine, but the grammatical errors (mostly related to capitalization), spelling ("Havill"), and formatting (scientific names should be italicized) need to be corrected.
Response: Thank you. The manuscript is edited to correct these errors.
Reviewer 3 Report
Comments and Suggestions for Authors
Overall this is a useful and well designed study which will be helpful in managing HWA. My main comment is with regard to the statistical analysis using a multi-level one-way ANOVA, instead of a factorial analysis. If you're going to use a one-way approach, the use orthogonal contrasts is the best way to extract the various effects, granted. However, why not break the analysis down by the factors which would then a clearer picture of main effects and, most importantly, interactions. You cannot do a full 3-way factorial because the adjuvant was not tested in all treatment combinations, but I suggest running a 2-way for type of application and rate of application. If results are no different than the current one-way, fine. But interactions could well be significant and easy to plot in a nice graph. You could add the time intervals post treatment as a repeated-measures factor to round out a complete analysis.
Some additional comments:
lines 178-186. necessary? You're basically looking for 2 compounds, imidacloprid and olefin. Keep it simple.
l. 203. km or miles?
l. 210. why time in seconds if $ rate is per hour?
l. 223. "4) cost comparisons" is not a contrast
Author Response
We thank the reviewers for providing valuable recommendations. Our revised manuscript reflects the recommended changes. We used track changes to highlight our additions. In this revision, we have updated the in-text citations to reflect the journal's preference for numerical listing of citations. We did not highlight these changes with track changes as it would be cumbersome to read the document.
We address specific comments below.
REVIEWER 3
Comments and Suggestions for Authors
Overall this is a useful and well designed study which will be helpful in managing HWA. My main comment is with regard to the statistical analysis using a multi-level one-way ANOVA, instead of a factorial analysis. If you're going to use a one-way approach, the use orthogonal contrasts is the best way to extract the various effects, granted. However, why not break the analysis down by the factors which would then a clearer picture of main effects and, most importantly, interactions. You cannot do a full 3-way factorial because the adjuvant was not tested in all treatment combinations, but I suggest running a 2-way for type of application and rate of application. If results are no different than the current one-way, fine. But interactions could well be significant and easy to plot in a nice graph. You could add the time intervals post treatment as a repeated-measures factor to round out a complete analysis.
Response: Thank you for this suggestion. We worked with our college statistician during the study design and analysis, who recommended orthogonal contrasts. We do not feel that conducting post-hoc analysis is appropriate.
Some additional comments:
lines 178-186. necessary? You're basically looking for 2 compounds, imidacloprid and olefin. Keep it simple.
Response: This paragraph has been deleted
- 203. km or miles?
Response: kilometers is correct
- 210. why time in seconds if $ rate is per hour?
Response: it was hours, thank you for catching this
- 223. "4) cost comparisons" is not a contrast
Response: correct, we have deleted “comparison”